# Understanding the Mechanisms of SiC–Water Reaction during Nanoscale Scratching without Chemical Reagents

**DOI:** 10.3390/mi13060930

**Published:** 2022-06-11

**Authors:** Zhihao Cheng, Qiufa Luo, Jing Lu, Zige Tian

**Affiliations:** 1Institute of Manufacturing Engineering, Huaqiao University, Xiamen 361021, China; 20013080008@stu.hqu.edu.cn (Z.C.); zigetian@stu.hqu.edu.cn (Z.T.); 2National & Local Joint Engineering Research Center for Intelligent Manufacturing Technology of Brittle Material Products, Xiamen 361021, China

**Keywords:** 4H-SiC wafer, amorphous silica, ReaxFF reactive molecular dynamics, plastic removal, chemical reaction

## Abstract

Microcracks inevitably appear on the SiC wafer surface during conventional thinning. It is generally believed that the damage-free surfaces obtained during chemical reactions are an effective means of inhibiting and eliminating microcracks. In our previous study, we found that SiC reacted with water (SiC–water reaction) to obtain a smooth surface. In this study, we analyzed the interfacial interaction mechanisms between a 4H-SiC wafer surface (000$\overset{-}{1}$) and diamond indenter during nanoscale scratching using distilled water and without using an acid–base etching solution. To this end, experiments and ReaxFF reactive molecular dynamics simulations were performed. The results showed that amorphous SiO_2_ was generated on the SiC surface under the repeated mechanical action of the diamond abrasive indenter during the nanoscale scratching process. The SiC–water reaction was mainly dependent on the load and contact state when the removal size of SiC was controlled at the nanoscale and the removal mode was controlled at the plastic stage, which was not significantly affected by temperature and speed. Therefore, the reaction between water and SiC on the wafer surface could be controlled by effectively regulating the load, speed, and contact area. Microcracks can be avoided, and damage-free thinning of SiC wafers can be achieved by controlling the SiC–water reaction on the SiC wafer surface.

## 1. Introduction

As regards semiconductor development, silicon carbide (SiC) materials are currently known to exhibit excellent material advantages such as high forbidden bandwidth, high thermal conductivity, and high electron migration rate as compared with silicon materials [1,2,3,4]. Therefore, SiC devices are being widely used in various fields such as new energy-efficient vehicles, smart grids, and aerospace applications [5,6,7]. However, under high-power and extreme working conditions, the self-heating effects of SiC semiconductor materials gradually become evident and lead to poor heat dissipation and performance degradation [8]. Currently, the most effective method to solve the problem of wafer heat dissipation is the wafer back-thinning process [9,10]. Generally, SiC is considered as a typical difficult-to-machine material owing to its high mechanical hardness and chemical inertness; wafer thinning processes are constantly being improved to solve these problems [11], resulting in SiC wafers that have been thinned to less than 100 µm [12]. Therefore, ultrathin wafers with excellent performance are of remarkable significance for high-power and extreme working conditions. Generally, the wafer back-thinning process for SiC wafer involves mechanical grinding using diamond abrasives, directly resulting in wafer fragmentations and deformations owing to the formation of microcracks [13,14].

To produce thin SiC wafers and to improve their surface quality, some researchers have used fixed diamond abrasive tools for the SiC thinning process. Yam et al. [15] used a #2000 diamond wheel to grind SiC under a constant pressure, and a surface roughness Ra of 1.57 nm was achieved. Tsukimoto et al. [16] found that a 2.4 µm subsurface damage (SSD) layer existed on the SiC wafer after grinding with a #2000 diamond cup wheel, and microcracks were generated during the hard abrasive interaction and related plastic deformation and fracture. Although conventional grinding methods have achieved grinding effects, a few microcracks and SSD are still generated owing to hard abrasives with unequal protrusion heights on the surface of the tool. To improve the surface quality of SiC wafers further, Feng et al. [17] proposed a polyvinyl alcohol/phenolic resin (PVA/PF) composite sol–gel diamond wheel based on traditional grinding and obtained a high surface quality of the 4H-SiC wafer. However, the pores on the surface of the sol-gel diamond wheel were easily blocked, and the diamond wheel needed to be dressed over time. Furthermore, a few unconventional methods have been proposed for SiC thinning. Sano et al. [18] proposed a method of plasma etching with high-pressure SF6 plasma to reduce the thickness of a 2-inch wafer, and the thickness of SiC wafer could be reduced to approximately 100 μm within 20 min of plasma etching. This technology could be used as an effective method for thinning SiC wafers; however, some etch pits were not completely removed. Guan et al. [19] proposed an electric discharge grinding (EDG) method for SiC wafer thinning, which could obtain a small-sized SiC wafer with a minimum thickness of 30 µm; however, the surface roughness and subsurface quality of EDG were lower than those of the other methods. Although unconventional methods can effectively obtain ultrathin chips, the current technology is not sufficiently developed, and SSD is still a serious concern. The problem of subsurface damage introduced by SiC thinning, thus, remains unsolved, regardless of whether conventional or unconventional methods are employed. Therefore, obtaining high-quality thin wafers is difficult. Moreover, to eliminate microcracks and improve surface quality, chemical mechanical polishing (CMP), which is an effective method to obtain smooth and damage-free wafers through chemical reactions, is employed; however, the material removal rate (MRR) of SiC wafers is significantly low because of the high hardness and stable chemical properties of the SiC materials. Therefore, the production of high-performance SiC wafers is limited because CMP using acid–base chemical reagents is expensive and polluting [20,21,22].

In our previous studies [23,24,25,26], a green-energy and efficient semi-fixed abrasive tool for ultra-precision machining, referred to as an SG pad, was proposed. A smooth, scratch-free, and nearly damage-free SiC wafer surface (SSD of only 2 nm) could be achieved using this tool. Instead of acid, alkali, or other active substances, only distilled water was used as the coolant in the SiC processing; this effectively mitigated environmental pollution. It was found that the SiC wafer could react with deionized water and produce amorphous silicon dioxide (SiO_2_) under mechanical scratching using diamond abrasives [27]. This technique could be used for the subsequent treatment of SiC thinning by eliminating microcracks and improving material removal rate via the utilization of the SiC–water reaction, which would avoid environmental pollution. Thus, SiC wafers can be obtained rapidly and efficiently by controlling the SiC–water reaction, and the conditions of the reaction should be further studied. In this regard, traditional molecular dynamics (MD) simulations based on classical force fields are used to analyze the contact behavior and material removal among atoms during nano-machining, but classical force fields cannot be used to describe reactive chemical systems [28,29]. ReaxFF reaction molecular dynamics (RMD) simulation based on the reactive force field approach can effectively simulate the formation and dissociation of chemical bonds [30,31,32,33], which has already been successfully applied to various processes based on the interaction of chemical reactions and mechanical effects, such as the interaction between water and Si [34]. Therefore, RMD simulation was used to investigate the interaction between water and SiC in this study. A method to induce a chemical reaction between SiC and water by mechanical scratching of SiC with a single diamond indenter was proposed to determine the conditions of the SiC–water reaction that can optimize the thinning process to satisfy the accuracy requirements for high-quality SiC wafers. This study aimed to regulate the scratch parameters (contact state, scratch speed, and scratch load) to control the plastic removal of 4H-SiC and realize the chemical reaction between SiC wafers and water by conducting relevant experiments and RMD simulations. The characterizations of the scratched area using transmission electron microscopy (TEM), high-resolution TEM (HRTEM), selected area electron diffraction (SAED), energy dispersive spectrometry (EDS), Raman spectroscopy, and X-ray photoelectron spectrometer (XPS) were performed after the scratching process. RMD simulations were performed to investigate the chemical reaction between the C-face (000$\overset{-}{1}$) of 4H-SiC and water molecules under the mechanical action of a diamond indenter during nanoscale scratching.

## 2. Materials and Methods

### 2.1. Experimental Method

A friction and abrasion tester (MFT-5000, Retc, Wilmington, MA, USA) was used for scratching the SiC wafer. The size of SiC (4H-SiC-4°, (000$\overset{-}{1}$) plane) wafers was 20 mm × 20 mm, and the original surface roughness Ra of SiC wafers was approximately 0.5 nm. The main factors of the SiC–water reaction in this experiment were determined using the control variable method, which controlled the radius of the indenter, scratching speed, and load—in that order. Diamond indenter tools were used for these experiments; these tools were made of single-crystal diamonds with radii of 0.2 mm, 0.4 mm, 0.6 mm, 1 mm, and 2 mm. The rotational speeds were set to 500 rpm, 1000 rpm, 1500 rpm, and 2000 rpm, and the eccentric radii were set to 5 mm, 6 mm, 7 mm, and 8 mm, corresponding to linear speeds of 0.262 m/s, 0.628 m/s, 1.099 m/s, and 1.625 m/s, respectively. The scratch loads were set to 0.3 N, 0.6 N, 0.9 N, 1.2 N, 1.5 N, and 1.8 N in these experiments. The processing time was set to 1 min. No chemical substance was added during the scratching process, and only distilled water was used as the coolant.

To better analyze the material removal behavior of SiC samples, the surface topographies of the scratches on SiC wafers were measured using a three-dimensional optical surface profiler (Newview 7300, ZYGO, Middlefield, CT, USA). Raman spectroscopy and XPS (K-alpha, Thermo Fisher, Waltham, MA, USA) were used to analyze the chemical composition of the scratches. TEM, HRTEM, SAED, and EDS characterizations of the wear debris extracted from the scratching-process coolant were processed using a Talos F200X G2 field-emission gun transmission electron microscope with an accelerating voltage of 200 kV.

### 2.2. RMD Model

The RMD simulation of scratching on the C-face of SiC with a diamond abrasive was performed to study the process of tribochemical reactions at nanoscale. The RMD simulation model of the high-speed scratching on SiC is shown in Figure 1; the workpiece size was 21.567 Å × 21.346 Å × 47.316 Å. The model was composed of a diamond abrasive tool, C-face (000$\overset{-}{1}$) of the 4H-SiC wafer, and water molecules. The top part of the model represented a simplified diamond abrasive, and the contact interface of the diamond abrasive was divided into planar, spherical, and conical surfaces to consider the effects of different contact states. The middle part of the model represented free water molecules, and the lower part of the model represented a standard 4H-SiC lattice. The top two layers of SiC atoms used to control the loading and friction of the diamond abrasive were set as the moving layers, and the function of the bottom two layers was fixed to prevent rigid movements of the entire structure during the friction process.

The accuracy and reliability of RMD simulations depend on the appropriate potential function selected. The ReaxFF approach was used to simulate bond breaking and bond formation during chemical reactions in the RMD simulations. Therefore, ReaxFF was used to describe the chemical reaction between SiC and H_2_O molecules in the friction process. Therefore, all simulations were performed in the NVT ensemble, and the time step was set to 0.25 fs. A periodic boundary condition was applied in the x- and y-directions to reduce the influence of boundary effects, and a fixed boundary condition was applied in the z-direction. The temperature was controlled using a Nose–Hoover thermostat; the initial temperature of the simulation system was 160 °C, and the damping constant was 10 fs. The RMD simulations in this study were performed using a large-scale atomic/molecular massively parallel simulator (LAMMPS). A massively parallel simulator (MAPS) was used to construct the model, visualize the process, and analyze the results.

## 3. Results

### 3.1. Controlling the Material Removal Behaviour of SiC Wafer

Different radii were set for the indenter, scratch speeds, and loads during this experiment to reflect the mechanical action of a single diamond abrasive in SiC wafer ultra-precision machining to examine the SiC–water reaction conditions. First, the same load (0.3 N) and linear velocity (0.262 m/s) were set, and the results of the C-face (000$\overset{-}{1}$) morphologies of 4H-SiC wafers scratched by diamond indenters having different radii are depicted in Figure 2. After scratching with a diamond indenter of 0.2 mm radius, a brittle-removal phenomenon was observed on this scratch (Figure 2a), as well as several deep pits and microcracks with micron grade depth. With increasing radius of the diamond indenter, the scratch depth decreased from micro-scale to nanoscale, and the brittle-removal mode of the SiC material gradually weakened. A scratch with no evident pits and cracks was achieved after scratching with a diamond indenter of 2 mm radius, as shown in Figure 2e, and the material removal mode of SiC changed to plastic removal.

Subsequently, a diamond indenter of 2 mm radius and a constant load of 0.3 N were controlled in this experiment, and the speeds were set sequentially to 0.262 m/s, 0.628 m/s, 1.099 m/s, and 1.625 m/s to study the effects of the scratching speed on SiC material removal. As shown in Figure 3, the depths and widths of scratches increased with an increase in speed. Furthermore, the shapes of scratches without any pits or cracks were more regular, and the extent of plastic removal of the SiC material gradually increased. Therefore, a speed of 1.625 m/s was set as a fixed parameter to deepen the extent of plastic removal of the SiC material while examining the SiC–water reaction. 

Finally, a diamond indenter of 2 mm radius and a constant speed of 1.625 m/s were fixed in this test, and the loads were set to 0.3 N, 0.6 N, 0.9 N, 1.2 N, 1.5 N, and 1.8 N to study the effects of different loads on SiC material removal. The depths and widths of the scratches were increased with the increase in load shown in Figure 4a–d and when the load was between 0.3 N and 1.2 N, the depth of scratches increased from 15 nm to over 130 nm, and the extent of plastic removal of SiC material increased. However, a few pits were observed at the bottom of the scratch, which were at a depth of 400 nm, as shown in Figure 4e. The coexistence of brittle removal and plastic removal were observed when the load was 1.5 N. Nevertheless, a few deep microcracks were observed on the scratch when the load was 1.8 N, as shown in Figure 4f, when the SiC material was primarily in the brittle stage. With the increase in load, there was a large amount of fragmentation at the bottom of the scratch, gradually transitioning from the plastic stage to the brittle–plastic coexistence stage, and subsequently to the brittle stage.

### 3.2. Characterisations of SiC–Water Reaction on C-Face (000$\overset{-}{1}$)

As indicated by results of the afore-described experiments, when the radius of diamond indenter was 2 mm, speed was 1.625 m/s, and load was 1.2 N, a relatively uniform and regular plastic scratch without evident cracks was obtained, and wear debris was collected from the SiC wafer surface after the experiment. Therefore, the plastic scratch on the SiC surface was ultrasonically cleaned and analyzed to investigate whether the SiC–water reaction was performed under plastic conditions using Raman spectroscopy and XPS. The wear debris was examined and analyzed to further study the chemical reaction that occurred between the C-face (000$\overset{-}{1}$) of SiC and water. The characterization results of the wear debris included TEM, HRTEM, SAED, and EDS analyses.

Figure 5 shows the Raman detection of the C-face (000$\overset{-}{1}$) of 4H-SiC wafer under the abovementioned plastic conditions. The Raman peak of SiC was observed at 966 cm^−1^, whereas the Raman peak of silica dioxide was at 968 cm^−1^ [35]. After fitting the original graph to split the peaks, new peaks of SiC and silica dioxide were at 963.6 cm^−1^ and 973.1 cm^−1^, respectively, as shown in Figure 5a. This is because the oxide layer remained on the original surface of the SiC wafer after CMP; therefore, a smaller peak of silica dioxide appeared. In Figure 5b, the peaks of SiC and SiO_2_ were at 970.1 cm^−1^ and 987.4 cm^−1^, respectively. The shift of the Raman peak of SiC may be because of the influence of stress [36]. Comparison of Raman results between scratch and non-scratch regions, it could be clearly seen that the original peak was significantly broadened, and the fitted SiO_2_ peak was also broadened, indicating that amorphous SiO_2_ was possibly formed on the scratch [37].

The composition and chemical state of the plastic scratch on the C-face (000$\overset{-}{1}$) of 4H-SiC wafers were further characterized by XPS, as shown in Figure 6. As observed in Figure 6a, there are mainly peaks at Si 2s, Si 2p, C 1s, N 1s, O 1s, and O KLL. The N 1s peak was attributed to nitrogen doping during the preparation of 4H-SiC wafers. The fine spectra of Si 2p, C 1s, and O 1s were analyzed because the main elements on the surface of 4H-SiC wafers were Si, C, and O. There were four peaks in the C 1s spectra; the peak at 284.78 eV corresponded to SiC, whereas the other three peaks corresponded to the organic compounds adsorbed on the surface. The peak at 534.31 eV in the O 1s spectra and the peak at 103.42 eV in the Si 2p spectra corresponded to SiO_2_ [38,39]. Therefore, SiO_2_ existed in the scratch on the C-face (000$\overset{-}{1}$) of 4H-SiC wafer, and the results indicated that the SiC might react with water under mechanical friction to generate SiO_2_. Further analysis of the wear debris was required to verify whether the SiC–water reaction had occurred. 

TEM images, SAED patterns, HRTEM images, EDS spectra, and EDS mapping of SiC wear debris under the aforementioned conditions are shown in Figure 7. As shown in Figure 7a,b, the wear debris of the C face with a thickness of approximately 200–300 nm was spherical, and the atomic arrangement of the wear debris had no periodic variation, with no evident crystalline state. The SAED pattern without diffraction spots emerged as a halo, indicating that the wear debris was in an amorphous phase. The Mo and Cu appearing in the EDS spectra were mainly derived from the Mo mesh micro-grid supporting film, as shown in Figure 7c. The elements of the wear debris were silicon, oxygen, and carbon; however, oxygen and silicon were considerably more abundant than carbon. In addition, among the individual elements observed in the EDS mapping shown in Figure 7d, the major elements of the wear debris are silicon and oxygen. Therefore, it was confirmed that the main component of the wear debris of the C-face was amorphous SiO_2_ [40,41]. 

### 3.3. Mechanism of SiC–Water Reaction in RMD Simulation

Figure 8 shows the results of the reaction between SiC and water molecules under different contact states. The diamond indenter had a planar surface, spherical surface, and conical surface. The friction load, friction speed, and temperature were set to 60 nN, 20 m/s, and 580 °C, respectively. From different viewpoints, the results showed that the lattice structure of SiC materials was severely damaged by the conical surface, and, thus, more silicon atoms were oxidized. In this case, the layer structure of SiC materials was damaged by the spherical contact surface, which reacted with water molecules during the friction process. However, the structure of SiC materials was relatively intact under planar contact, with only the surface layer atoms forming silicon–oxygen bonds with oxygen atoms. This further indicated that the degree of chemical reaction at the interface was determined by the degree of damage to the lattice structure of SiC materials. Interfacial chemical reactions could be facilitated by contact modes that resulted in greater stress concentrations, leading to structural breakdown of SiC materials. Generally, to match the actual processing situation, the contact mode at the nanoscale was simplified to planar contact. Therefore, all simulations were performed using planar contact to investigate the effects of the remaining factors on the SiC–water reaction.

The results of the reaction between SiC and water molecules at different friction speeds are shown in Figure 9, wherein the friction depths at a temperature of 790 °C were 0.6 nm and 0.7 nm. When the speed increased from 10 m/s to 20 m/s, the lattice structure of SiC materials did not significantly change at the same friction depth. Conversely, at the same friction speed, when the friction depth increased from 0.6 nm to 0.7 nm, the lattice structure of SiC materials was significantly damaged, effectively promoting the SiC–water reaction. The change in friction speed had no significant effect on the chemical reaction at the interface between SiC and water molecules, as well as on the degree of damage to the lattice structure of SiC materials.

The SiC–water reactions at different temperatures are shown in Figure 10, wherein the friction depths at a friction speed of 20 m/s were 0.6 nm and 0.7 nm. From the perspective of temperature, at the same friction depth, when the temperature was below 580 °C, the interfacial reaction only occurred at the contact interface between SiC materials and water molecules. At temperatures between 580 °C and 790 °C, the water molecules entered the second layer of SiC materials, starting the chemical reaction. When the temperature was further increased to 1000 °C, the lattice structure of SiC materials was significantly damaged, and the reaction occurred in the subsurface layer. As the temperature increased, the atoms at the interface moved more drastically, and the chemical reactions that occurred became more obvious.

From the point of view of friction depth (load), when the friction depth increased from 0.6 nm to 0.7 nm at the same temperature, the lattice structure of SiC materials was significantly damaged, and as the friction depth increased, the extent of SiC lattice structure damage increased and the reaction became more vigorous. Simultaneously, even at lower temperatures, as long as the friction depth was sufficiently large, the lattice structure of SiC materials would be damaged for the reaction to occur. Notably, as shown in Figure 10a, the lattice structure of SiC materials was still not significantly damaged at a temperature of 790 °C and friction depth of 0.6 nm; however, the lattice structure was already significantly disrupted at a temperature of 370 °C and friction depth of 0.7 nm, resulting in the SiC–water reaction. Therefore, the lattice structure of SiC materials was more effectively disrupted by increasing the friction depth than by increasing the temperature; thus, the reaction at the interface was significantly promoted.

## 4. Discussion

A regular SiC plastic scratch without evident cracks was obtained in the scratching experiments that employed a diamond indenter with a radius of 2 mm, speed of 1.625 m/s, and load of 1.2 N. In this process, only deionized water was used as the coolant without adding any acid–base etching solution. The chemical composition of the SiC plastic scratch was further analyzed using Raman spectroscopy and XPS, and the wear debris was analyzed using TEM. The chemical reaction was observed to generate amorphous silicon dioxide (the phenomenon of the chemical reaction that occurred between the diamond abrasive and SiC wafers was highly consistent with our previous work [27]). This result was highly consistent with the results of RMD simulations, which further verified that the chemical reaction at the nanoscale between SiC and water molecules occurred under mechanical action. The experimental results indicate that the SiC–water reaction occurred in the plastic removal stage of SiC wafers; thus, controlling the material removal of SiC was a prerequisite for determining the SiC–water reaction. Scratching experiments focused on the removal mode of SiC to indirectly verify the conditions of the SiC–water reaction at the macroscopic level, whereas the scratching depths at the nanoscale were controlled by precisely regulating the scratch loads, which was consistent with the scale of friction depths at the nanoscale in the RMD simulation. Therefore, both experiments and RMD simulations were combined to investigate the effects of different contact states, friction speed, friction depth (load), and different temperatures on the SiC–water reaction.

In the experiments, the brittle fracture of bulk material on the SiC surface was directly caused by the small radius indenter (a radius of 0.2 mm), indicating that the removal mode of SiC materials in this state was brittle. However, the material removal mode of SiC was plastic when SiC was scratched using a large-radius indenter (with a radius of 2 mm). In the RMD simulation, the lattice structure of SiC materials was easily damaged by the conical and spherical contacts; however, in the case of the planar contact, the lattice structure was relatively more intact, and the reaction occurred on the surface. Both the experiments and RMD simulations further illustrated that the structure of SiC was easily damaged owing to a larger stress concentration that occurred because of the difference in contact states, and the material removal mode was more likely to be brittle. Therefore, the SiC–water reaction can be effectively promoted only if the material removal form of SiC is controlled in the plastic removal phase.

As the scratching speed gradually increased (up to 1.625 m/s), the extent of plastic removal of SiC materials gradually increased during the experiment, and the plastic scratches became increasingly evident. Nevertheless, the actual experimental speed was not simulated, and the mode of material removal was less affected by high speeds in the RMD simulation. Hence, the agreement between the experimental and simulated results verified that the scratching speed had no evident effect on the material removal of SiC materials and the SiC–water reaction, whereas it was easier to observe plastic scratches with an increase in speed [42].

The load was precisely controlled within a certain range (0.3–1.5 N) in the experiment, which precisely controlled the friction depth. When the load was 1.2 N, a regular plastic scratch without any evident cracks was obtained, and the SiC–water reaction was confirmed to occur on the scratch. The SiC material changed from plastic deformation mode to brittle damage mode with an increase in load, whereas brittle cracks appeared at the bottom of the scratch when the load was greater than 1.5 N. Therefore, the brittle–plastic transformation behavior of SiC was realized in the range of 1.5–1.8 N. An increase in the friction depth (load) in the RMD simulation increased the extent of SiC lattice structure damage, and the interface reaction became significantly vigorous. Generally, the higher the temperature, the more drastic the chemical reaction is. Although it was difficult to conduct high-temperature scratching experiments owing to the limitations of the instruments and aqueous environment, RMD simulation was performed to analyze the effect of different temperatures on the SiC–water reaction. Evidently, with an increase in temperature, the damage in the lattice structure of SiC gradually increased. However, by combining the two conditions of friction depth (load) and temperature, even at a lower temperature (160 °C), the friction depth was sufficiently large; this damages the lattice structure of SiC and promotes the SiC–water reaction. This result was verified by the scratching experiment, wherein the SiC–water reaction still occurred at room temperature (20 °C) under the abovementioned conditions. Therefore, the friction depth (load) during the plastic removal stage was more important than the temperature in the mechanism of the SiC–water reaction; this indicates that the load was the most critical factor in achieving plastic removal of SiC in both experiments and RMD simulations. By effectively regulating the contact state, speed, and load, the material removal mode of SiC materials was reasonably controlled in the plastic removal phase; thus, the SiC–water reaction smoothly occurred.

The SiC-water reaction occurred under the plastic removal of SiC materials, thus controlling the reaction was primarily a matter of controlling the mode of material removal in an aqueous medium. The removal mode of SiC was achieved primarily by controlling the load and contact states, and secondarily by controlling the velocity and temperature. The lattice structure of SiC materials was directly and effectively damaged by regulating the load and contact states in an aqueous medium. Thus, the material removal form of SiC was effectively controlled in the plastic removal phase, further avoiding microcracks and brittle collapse. During the plastic phase, SiC materials on surface layer was mechanically induced to generate amorphous silicon dioxide when distilled water was used as the coolant, and a controllable SiC–water reaction was used to obtain smooth, scratch-free, and damage-free wafers during the thinning process of SiC wafers.

## 5. Conclusions

This study entailed experiments and RMD simulations to investigate the mechanisms of the reaction between SiC and water during nanoscale scratching in the absence of any chemical reagents. The material removal behavior of SiC at the nanoscale level was analyzed, and the mechanisms of the SiC–water reaction were summarized. The following conclusions were drawn:(1)A regular plastic scratch with nanometer depth and almost no evident cracks was obtained using diamond abrasives to scratch 4H-SiC C face (000$\overset{-}{1}$) without any chemical agent, and the formation of amorphous SiO_2_ was clearly observed, which verified that the SiC–water reaction occurred in the SiC plastic removal stage.(2)When the contact radius was gradually increased to 2 mm, the scratches that developed on the surface were converted from brittle to plastic removal mode, which were further concluded from the RMD simulations. The SiC crystal structure was damaged to a lesser extent owing to the larger contact radius and smaller contact stress. The plastic removal of SiC materials was not significantly affected by the increase in speed; only the removal rate of SiC materials was increased.(3)With the increase in load, the scratch depth could be in the range of tens to hundreds of nanometers during scratching. Furthermore, the removal mode of SiC transitioned from plastic to brittle and thereby achieved a controllable behavior in the plastic domain of SiC under controlled loading, which indirectly controlled the SiC–water reaction. Conversely, the controllable behavior of the SiC–water reaction at the nanoscale during RMD simulations was primarily achieved by adjusting the friction depth. Therefore, both experiments and simulations regulated the load, thereby regulating the friction depth to control the occurrence of the SiC–water reaction.(4)The main factors controlling the SiC–water reaction were the load and contact states, and the secondary factors were the speed and temperature. The SiC–water reaction could occur smoothly at low temperatures as long as a suitable load and contact state were maintained. Therefore, microcracks could be reasonably avoided by the effective application of the SiC–water reaction in the subsequent SiC thinning process, thus improving the surface quality of SiC wafers.

## Figures and Tables

**Figure 1 micromachines-13-00930-f001:**
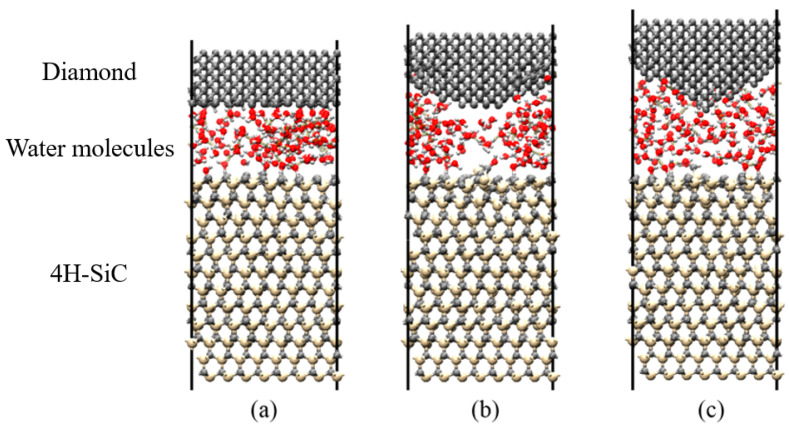
ReaxFF reactive molecular dynamics (RMD) simulation with (**a**) planar, (**b**) spherical, and (**c**) conical contact.

**Figure 2 micromachines-13-00930-f002:**
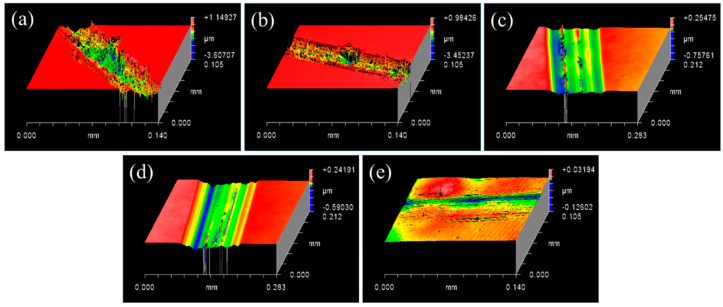
Surface profile of SiC scratched with 0.3 N, 0.262 m/s by different radius of indenter (**a**) 0.2 mm, (**b**) 0.4 mm, (**c**) 0.6 mm, (**d**) 1 mm, and (**e**) 2 mm.

**Figure 3 micromachines-13-00930-f003:**
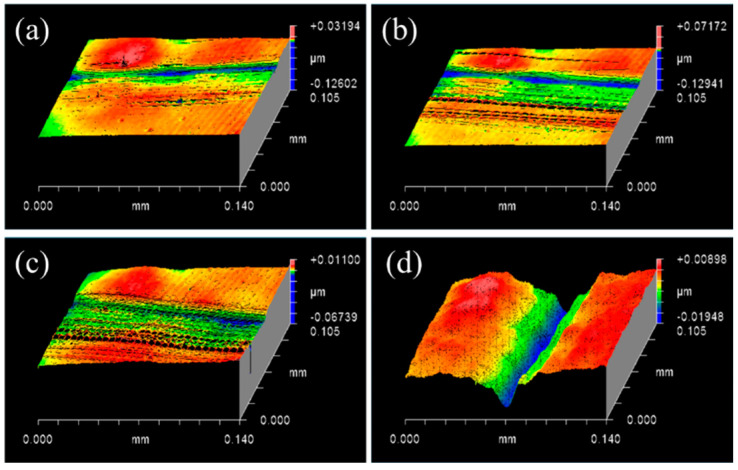
Surface profiles of scratched SiC after experiments with 2 mm indenter and 0.3 N at different speeds: (**a**) 0.262 m/s, (**b**) 0.628 m/s, (**c**) 1.099 m/s, and (**d**) 1.625 m/s.

**Figure 4 micromachines-13-00930-f004:**
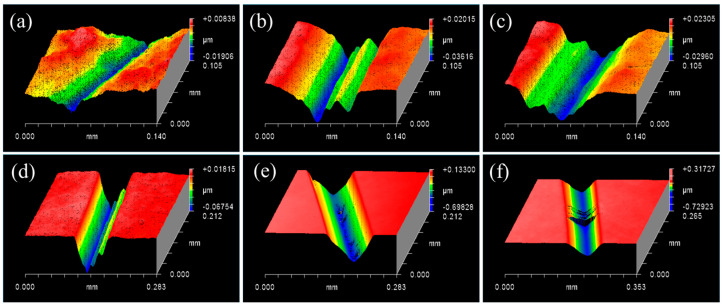
Scratch surface profiles of SiC scratched with a 2 mm indenter and speed of 1.625 m/s at different loads: (**a**) 0.3 N, (**b**) 0.6 N, (**c**) 0.9 N, (**d**) 1.2 N, (**e**) 1.5 N, and (**f**) 1.8 N.

**Figure 5 micromachines-13-00930-f005:**
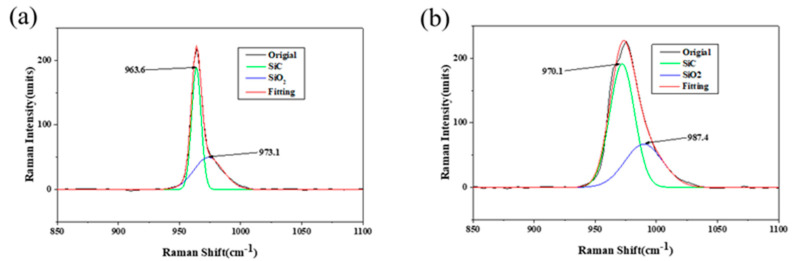
Detection results of debris of C-face (000$\overset{-}{1}$) of 4H-SiC wafer after scratching under optimal conditions: (**a**) Raman detection without scratching and (**b**) Raman detection with scratching.

**Figure 6 micromachines-13-00930-f006:**
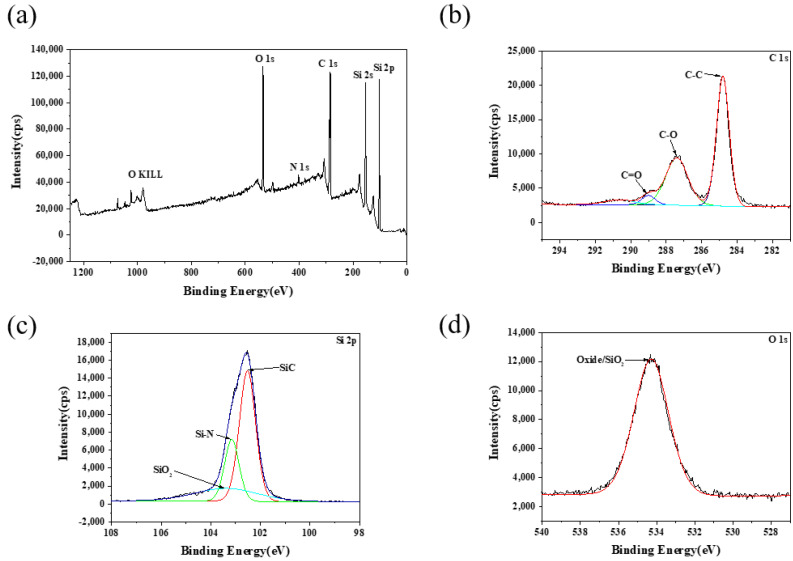
Detection results of debris of C-face (000$\overset{-}{1}$) of 4H-SiC wafer 4H-SiC wafers after scratching under optimal conditions: (**a**) XPS whole spectra, (**b**) XPS fine spectra of C 1s, (**c**) XPS fine spectra of Si 2p, and (**d**) XPS fine spectra of O 1s.

**Figure 7 micromachines-13-00930-f007:**
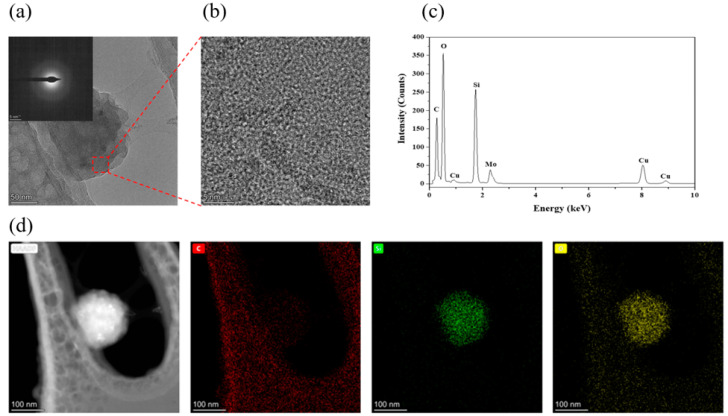
Detection results of debris of 4H-SiC wafers after scratching under optimal conditions: (**a**) TEM image and SAED pattern, (**b**) HRTEM image, (**c**) EDS spectra, and (**d**) EDS Mapping.

**Figure 8 micromachines-13-00930-f008:**
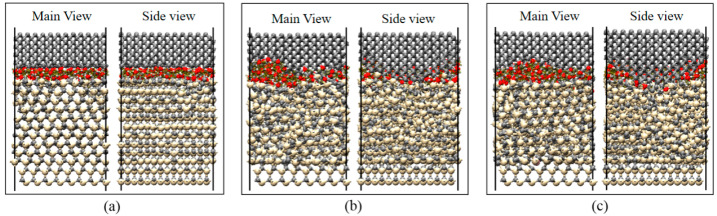
SiC–water reactions under different contact states when the contact interfaces were set as: (**a**) flat, (**b**) spherical, and (**c**) conical. Frictional loads, frictional velocities, and temperatures are shown as 60 nN, 20 m/s, and 580 °C.

**Figure 9 micromachines-13-00930-f009:**
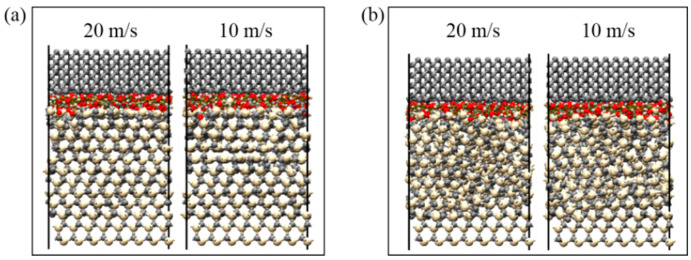
SiC–water reactions at different friction speeds. Friction depths and temperatures were set to: (**a**) 0.6 nm, 790 °C and (**b**) 0.7 nm, 790 °C.

**Figure 10 micromachines-13-00930-f010:**
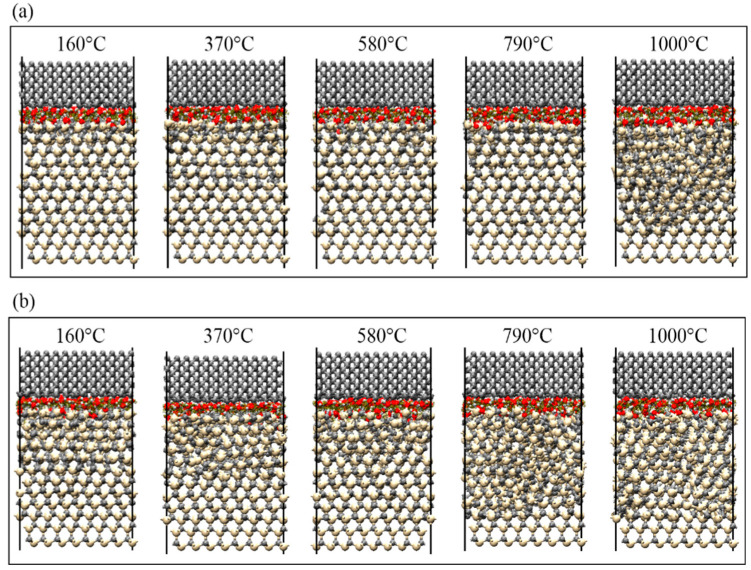
SiC–water reactions at different temperatures. The friction depth and friction velocity were set to: (**a**) 0.6 nm, 20 m/s and (**b**) 0.7 nm, 20 m/s.

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
