# Peer review of "Understanding the Mechanisms of SiC–Water Reaction during Nanoscale Scratching without Chemical Reagents"

_micromachines, 2022, doi:10.3390/mi13060930_

Round 1
Reviewer 1 Report
In this manuscript, the authors have investigated the physical mechanism of SiC-wafer reaction during nanoscale scratching without chemical reagents. The experimental data and simulation comparisons are relatively detailed, and it is acceptable if the present manuscript is modified according to the following remarks:
1. Some typos and grammatical errors should be corrected in the manuscript (For instance: " SiC wafers involves", " the extent of the plastic removal ", " from micrometres to nanometres",...etc). Thus, it is strongly recommended to conduct a careful revision of the whole manuscript.
2. Without the comparative experiment or simulation process using chemical reagents, the advantages of not using chemical reagents proposed by the author are not obvious enough. Therefore, I suggest that the authors can supplement the relevant comparison chart of chemical reagents, and ensure that the experimental conditions are consistent with the optimal experimental conditions explored by the authors
3. Lines and colors are unclear in Figure 5, no obvious contrast can be seen.
Author Response
Dear Editor,
Thank you for your letter and your useful comments concerning our manuscript entitled “Understanding the mechanisms of SiC–water reaction during nanoscale scratching without chemical reagents” (Manuscript ID: micromachines-1762850). Those comments are all valuable and very helpful for revising and improving our paper. We have studied comments carefully and have made corrections which we hope meet with approval. Revised portion are marked in red in the paper. The main corrections in the paper and the responds to the reviewer’s comments are as following:
Responds to the reviewer’s comments:
Reviewer: 1
- Some typos and grammatical errors should be corrected in the manuscript (For instance: "SiC wafers involves", "the extent of the plastic removal ", "from micrometres to nanometres", etc). Thus, it is strongly recommended to conduct a careful revision of the whole manuscript.
Response: We have revised the errors in the manuscript which are marked in red in the paper. ( For instance: " SiC wafers involves" has changed to " SiC wafer involves", " the extent of the plastic removal " has changed to " the extent of plastic removal ", " from micrometres to nanometres" has changed to " from micro-scale to nano-scale " etc) Also we have made some other changes in the manuscript.
- Without the comparative experiment or simulation process using chemical reagents, the advantages of not using chemical reagents proposed by the author are not obvious enough. Therefore, I suggest that the authors can supplement the relevant comparison chart of chemical reagents, and ensure that the experimental conditions are consistent with the optimal experimental conditions explored by the authors:
Response: Considering the reviewer’s suggestion, we have added relevant contents and highlighted the advantages of not using acid-base chemical reagents, which are marked in red in the “Introduction” part.
- Lines and colors are unclear in Figure 5, no obvious contrast can be seen.
Response: Considering the reviewer’s suggestion, we have redrawn the Figure 5 and highlighted the lines and colors of the peak around 960 cm-1 before and after scratching. The oxide layer remains on the original surface of SiC wafers after CMP, therefore a smaller peak of SiO2 was found in Figure 5 a. Comparison of Raman results between scratch and non-scratch regions, it could be clearly seen that the original peak was significantly broadened, and the fitted SiO2 peak was also broadened, indicating that amorphous SiO2 may be generated in the scratch area. We have added relevant contents, which are marked in red in the “Results” part.
Figure 5. Detection results of debris of C-face (0001(-)) of 4H-SiC wafer after scratching under optimal conditions: (a) Raman detection without scratching and (b) Raman detection with scratching.
Also we have made some other changes in the manuscript. These changes will not influence the content and framework of the paper. And here we did not list the changes but marked in red in revised paper.
We appreciate for Editors/Reviewers’ warm work earnestly, and hope that the correction will meet with approval.
Yours sincerely,
Qiufa Luo

Reviewer 2 Report
This paper presents a detailed experimental and theoretical investigation of the processes of surface oxidation during scratching off 4H-SiC surfaces. The importance of this study is caused by the possibility of the minimization of the formation of deep cracks applying mechanical polishing of water-covered surfaces. I find the presentation of the paper interesting, adequate, reliable and comprehensive. Obtained results will be definitely useful for the development of improved technologies of wafer polishing. Molecular dynamics simulations enable to disclose the pathways of the interaction of water with Si atoms in the surface atomic layers and are fully justified therefore. The material contained in the paper deserves to be published.
However, a number of issues should be addressed prior to the publication of this paper:
1. The quality of English language must be significantly improved.
2. The authors made investigations only for C-face (0001) surfaces of SiC crystal. Why? How would the results change if Si-face surface was considered?
3. Too few words were said about the choice of potential. To my opinion, just to mention that the ReaxFF potential is generally used to simulate chemical reactions by molecular dynamics simulations is too insufficient. Authors should better describe the advantages of this potential in comparison to other ones to convince the reader that this is the best choice.
4. I did not find any indication of the reasons why surface oxidation by water during mechanical removal of material improves surface quality and enables to avoid crack formation. To my opinion, this point should be discussed in the Discussion section.
After modification with respect to these comments, the paper may be published in the Micromachines journal.
Author Response
Dear Editor,
Thank you for your letter and your useful comments concerning our manuscript entitled “Understanding the mechanisms of SiC–water reaction during nanoscale scratching without chemical reagents” (Manuscript ID: micromachines-1762850). Those comments are all valuable and very helpful for revising and improving our paper. We have studied comments carefully and have made corrections which we hope meet with approval. Revised portion are marked in red in the paper. The main corrections in the paper and the responds to the reviewer’s comments are as following:
Responds to the reviewer’s comments:
Reviewer: 2
- The quality of English language must be significantly improved.
Response: Thank you for your valuable and thoughtful comments. We have carefully checked and found a professional editing agency to revise the article, and the editing report is shown in the figure.
- The authors made investigations only for C-face (0001) surfaces of SiC crystal. Why? How would the results change if Si-face surface was considered?
Response: SiC wafer thinning is to thin the C-face (0001(_)), but the Si-face (0001) is mainly used for fabricating devices. We have found the phenomenon of the SiC-water reaction occurred on both the C-face and Si-face during the SiC polishing processes in our previous works. Therefore, the results would not be changed if the Si face was considered.
- Too few words were said about the choice of potential. To my opinion, just to mention that the ReaxFF potential is generally used to simulate chemical reactions by molecular dynamics simulations is too insufficient. Authors should better describe the advantages of this potential in comparison to other ones to convince the reader that this is the best choice.
Response: Traditional molecular dynamics (MD) simulations based on classical force fields are used to analyze the contact behavior and material removal among atoms during nano-machining, but classical force fields cannot be used to describe reactive chemical systems [28-29]. ReaxFF reaction molecular dynamics (RMD) simulation based on the reactive force field approach can effectively simulate the formation and dissociation of chemical bonds [30-33], which has already been successfully applied to various processes based on the interaction of chemical reactions and mechanical effects, such as the interaction between water and Si [34]. Therefore, RMD simulation was used to investigate the interaction between water and SiC in this study. Considering the reviewer’s suggestion, we have compared the difference between MD and RMD simulation and listed the previous studies to highlight the advantages of using ReaxFF potential, which are marked in red in the “Introduction” part.
- Chrobak, D.; Tymiak, N.; Beaber, A.; Ugurlu, O.; Gerberich, W. W.; Nowak, R. Deconfinement leads to changes in the nanoscale plasticity of silicon. Nat. nanotechnol. 2011, 6(8), 480-484.
- Wu, Z.; Liu, W.; Zhang, L. Revealing the deformation mechanisms of 6H-silicon carbide under nano-cutting. Comput. Mater. Sci. 2017, 137, 282-288.
- Van Duin, A.C.; Dasgupta, S.; Lorant, F.; Goddard, W.A. ReaxFF: a reactive force field for hydrocarbons. J. Phys. Chem. A 2001, 105(41), 9396-9409.
- Chenoweth, K.; Van Duin, A.C.; Goddard, W.A. ReaxFF reactive force field for molecular dynamics simulations of hy-drocarbon oxidation. J. Phys. Chem. A 2008, 112(5), 1040-1053.
- Senftle, T.P.; Hong, S.; Islam, M.M.; Kylasa, S.B.; Zheng, Y.; Shin, Y.K.; Van Duin, A.C. The ReaxFF reactive force-field: development, applications and future directions. NPJ Comput. Mater. 2016, 2(1), 1-14.
- Wen, J.; Ma, T.; Zhang, W.; Psofogiannakis, G.; van Duin, A. C.; Chen, L.; Lu, X. Atomic insight into tribochemical wear mechanism of silicon at the Si/SiO2 interface in aqueous environment: Molecular dynamics simulations using ReaxFF reactive force field. Appl. Surf. Sci. 2016, 390, 216-223.
- Fogarty, J. C.; Aktulga, H. M.; Grama, A. Y.; Van Duin, A. C.; Pandit, S. A. A reactive molecular dynamics simulation of the silica-water interface. J. Chem. Phys. 2010, 132(17), 174704.
- I did not find any indication of the reasons why surface oxidation by water during mechanical removal of material improves surface quality and enables to avoid crack formation. To my opinion, this point should be discussed in the Discussion section.
Response: We have found that the SiC-water reaction occurred under the plastic removal of SiC materials, and microcracks was only appeared during the brittle removal stage. Therefore, controlling the chemical reaction was to control the removal form of SiC material in the plastic removal stage, it will furtherly avoid microcracks and improve surface quality. We have added relevant contents which are marked in red in the “Discussion” part.
Also we have made some other changes in the manuscript. These changes will not influence the content and framework of the paper. And here we did not list the changes but marked in red in revised paper.
We appreciate for Editors/Reviewers’ warm work earnestly, and hope that the correction will meet with approval.
Yours sincerely,
Qiufa Luo
